# Pollination service provided by honey bees to buzz-pollinated crops in the Neotropics

**Franklin H. Rocha[1], Daniel N. Peraza[1], Salvador Medina[2], José Javier G. Quezada-Euán** [1]*

1 Departamento de Apicultura Tropical, Campus Ciencias Biológicas y Agropecuarias- Universidad Autónoma de Yucatán, Mérida-Xmatkuil, México, 2 Facultad de Matemáticas-Campus de Ingenierías y Ciencias Exactas, Universidad Autónoma de Yucatán, Anillo Periférico, Mérida, México

* javier.quezada@correo.uady.mx

**Data Availability Statement:** the data are all contained within the manuscript and/or Supporting Information files.

**Funding:** To CONACyT-SADER project 291333 "Manejo sustentable de polinizadores" to JJGQE

## Abstract

Generalist honey bees grant significant pollination services worldwide. Although honey bees can provide compensatory pollination services, their service to buzz-pollinated crops, compared to specialized pollinators, is not clear. In this study, we assessed the contribution of Africanized honey bees (AHB) and native sonicating bees (NBZ) to the pollination of eggplant (*Solanum melongena)* and annatto (*Bixa orellana*) in Yucatan, Mexico, one of the largest producers of these crops in the Americas and a region with one of the largest densities of honey bees in the world. We first compared the relative frequency and abundance of both bee types on flowers of both crops. Secondly, we controlled access to flowers to compare the number and weight of fruit and number of seed produced after single visits of AHB and native bees. For a better assessment of pollination services, we evaluated the productivity of individual flowers multiply visited by AHB. The results were compared against treatments using pollinator-excluded flowers and flowers that were supplied with additional pollen, which allowed an overall measure of pollination service provision (PSP). Our results showed that AHB were the predominant flower visitors in both crops and that were poorly efficient on individual visits. Notably, fruit quantity and seed number increased concomitantly with the number of AHB visits *per* flower on eggplant, but not on annatto. Estimation of PSP revealed no pollination deficit on eggplant but that a deficit existed on the pollination services to annatto. We found that AHB numerical predominance compensates their poor individual performance and can complement the services of native bees on eggplant, but not on annatto. We discuss possible explanations and implications of these results for buzz-pollinated crops in the neotropics an area with little assessment of pollination services and a high density of honey bees.

## Introduction

Animal mediated pollination is basic to many economically important crops worldwide, so better understanding the effect of different floral visitors is fundamental for a sustainable agriculture [1]. Few studies have evaluated pollination services in the Neotropics [2–4]; consequently,

for supporting this research and CONACyT study grants to FHR and DNP. The funders had no role in study design, data collection and analysis, decision to publish, or preparation of the manuscript.

**Competing interests:** The authors have declared that no competing interests exist.

little is known about the pollination requirements of many crops and the relevance of different pollinators [3,5]. Important food crops of the Solanaceae family, like tomatoes and peppers, are of neotropical origin. Noteworthy, these and other major crops, like eggplant, kiwi, and blueberry are buzz-pollinated, a type of specialized pollination that requires sonication (vibration) of their poricidal anthers for pollen release prior to fertilization [6]. Nonetheless, the economic importance of buzz-pollinators has been little assessed in agricultural systems [6]. In the case of buzz-pollinated crops, bee species capable of sonication are more efficient pollinators on individual visits compared with species that cannot sonicate [7]. However, traits of different nature can affect the outcome of the interaction among flower visitors with specific plants, which could either improve or reduce the effect of the pollination services [8–10]. Thus, recognizing which insect groups are best suited to pollinate specific crops and how they interact for optimizing yield and quality, is essential for crop management.

The native bee fauna of Mexico is in the range of 1900 species, but the Yucatan Peninsula is a relatively poor area [11]. In a study conducted in altered landscapes of the Yucatan, 80 native bee species were found, being the family Apidae the most abundant and varied [12]. In contrast, although non-native to the Americas, honey bees have established thriving populations across Mexico [4,5,13]. In the Yucatan Peninsula managed and wild colonies of honey bees are found at some of its highest densities elsewhere, where hybrid Africanized honey bees (AHB) are predominant [14]. Across the globe, the majority of pollination services rely on the honey bee (*Apis mellifera*), which is the most commercially used pollinator [15]. In spite of their high abundance, the impact of AHB on crop production remains one aspect little studied [16]. Honey bees are generalist floral visitors that can increase the productivity of economically important crops [17]. However, they cannot vibrate their bodies to remove pollen from buzz-pollinated flowers [6] and they may outcompete other pollinators [18,19] which may compromise the reproductive success of plants requiring specialized pollination [15,20,21]. However, to date, no assessment has been conducted on the pollination service provided by AHB to buzz-pollinated crops [22]. As AHB are highly abundant in the Neotropics [23], their large perennial populations could potentially provide resilient pollination services and stability to crop yield, thus, assessing their contribution to buzz-pollinated crops is important for sustainable agriculture management [3,18,24,25].

The Solanaceae is one example of buzz-pollinated plant families that encompass many economically important crops, especially in the genus *Solanum* [6]. The African-Asian native eggplant (*S. melongena* Linnaeus, 1753), is a major crop in many countries whose pollination requirements remain poorly studied [26,27]. Eggplant flowers are hermaphroditic and, can be self-compatible [27], though heavier fruit and larger number of seeds can result from cross pollination by sonicating bees [28,29]. Mexico is a center of diversification of the genus *Solanum* and the country is one of the largest producers of *Solanum* crops in the Americas, including eggplant, with ca. 180,000 tons annually [30]. The Yucatan Peninsula is the second eggplant producer region in Mexico [31].

Annatto *Bixa orellana* Linnaeus, 1753 (also known as 'achiote') is another buzz-pollinated Neotropical tree in the family Bixaceae [32]. For annatto, seed number is most relevant for the productivity of the crop as this is the part of the fruit used as paste seasoning base in cuisine and the obtention of bixin a major apocarotenoid used as natural colorant. Yucatan is the largest producer (252 tons representing 44% of the national production) and consumer of achiote in Mexico [5]. Annatto produces perfect flowers and is self-compatible, but higher fruit set has been registered after cross-pollination by bees [33].

In our investigation we asked if generalist honey bees could provide adequate pollination services to economically important buzz-pollinated crops in Mexico. To answer that question, we first studied the frequency of AHB and native bees on eggplant and annatto in Yucatán,

Mexico. We then compared the efficiency of both types of bee comparing the production of fruit and seed after individual visits. For a better assessment of pollination service, we introduced treatments to analyze the effect of different number of AHB visits on flowers of both crops. Finally, we estimated pollination service provision (PSP) using fruit number and weight as well as seed production, as indicator of pollination deficit [4].

## Materials and methods

The study on eggplant was conducted in the locality of Yaaxhón (20˚ 15' 17.0", 89˚ 29' 37.4") between August and December 2019. The plot of approximately 0.3 ha had ca 2,500 plants irrigated by a dripper system. The experimental plot was organized in rows of ca 80 m. long and a distance of 1.5m x 1 m between plants.

The study on annatto was conducted in the locality of Xohuayán (20˚ 11' 5.18", 89˚ 23' 46.8"), between October 2020 to November 2021. The plot represented 1 ha with a total of 840 plants without irrigation system nor use of agrochemicals. The experimental plot was organized in rows of ca 125 m long and a distance of 4 m x 3 m between plants. The owner of both plots agreed on the study, allowed sampling and conducting of the experiments and no special permit was required.

### Frequency and behavior of floral visitors

**Eggplant.** We recorded the frequency of AHB and native bees on flowers. During a 30 min walk, 30 plants on one row were visually screened. One observer with experience in the identification of bees recorded the different types of bee present on the flowers. The surveys were conducted on five separate days, every two hours between 08.00 and 16.00 h, which covers the period of more abundant flower visitors and stigma receptivity of eggplant flowers [29,34]. The time spent in data registration was compensated, so that an effective 30 min observation period was conducted per survey. Thus, daily observations were performed for two and a half hours, accumulating a total of 12.5 h of observation for this crop. The ability to sonicate of the different bee species was determined by means of the characteristic buzzing sound they do while visiting the flowers [35], which was later corroborated by video recordings. During the surveys, we also used two digital video cameras with high color resolution (Sony-HDR-CX405), to corroborate the behavior of AHB and native bees when visiting the flowers, especially the mechanism for the removal of pollen. Each camera was placed on a support and positioned in front of a flower (focal sampling) for 15 min, resulting in a total of 12.5 h of recordings (n = 50).

For both crops, the identity of floral visitors was confirmed (to genus) by comparison with specimens at the bee collection of the CCBA-Universidad Autónoma de Yucatán. Voucher specimens from each bee genera (three to ten specimens) were also deposited at the same collection.

We first plotted the daily average number of specimens of each bee genera collected on eggplant (Fig 1A). This showed that AHB were the dominant floral visitors with approximately 60% of all recorded visits (Fig 1C). Non-sonicating stingless bees of the genera *Nannotrigona* and *Frieseomelitta* were also present, representing around 10% of all visitors. Among sonicating bees, the most abundant with ca. 20% of all floral visits were the small Halictid bees of the genera *Lasioglossum* and *Augochlora*. Large sonicating bees of the genera *Euglossa*, *Xylocopa* and *Eulaema* represented ca. 10% of all floral visitors. For a subsequent analysis we separated the specimens into three functional groups, non-sonicating (AHB) and native bees (NB) and sonicating native bees (NBZ). However, for the pollination efficiency assessment we did not include the stingless bees (*Nannotrigona* and *Frieseomelitta*) in a separate non-sonicating

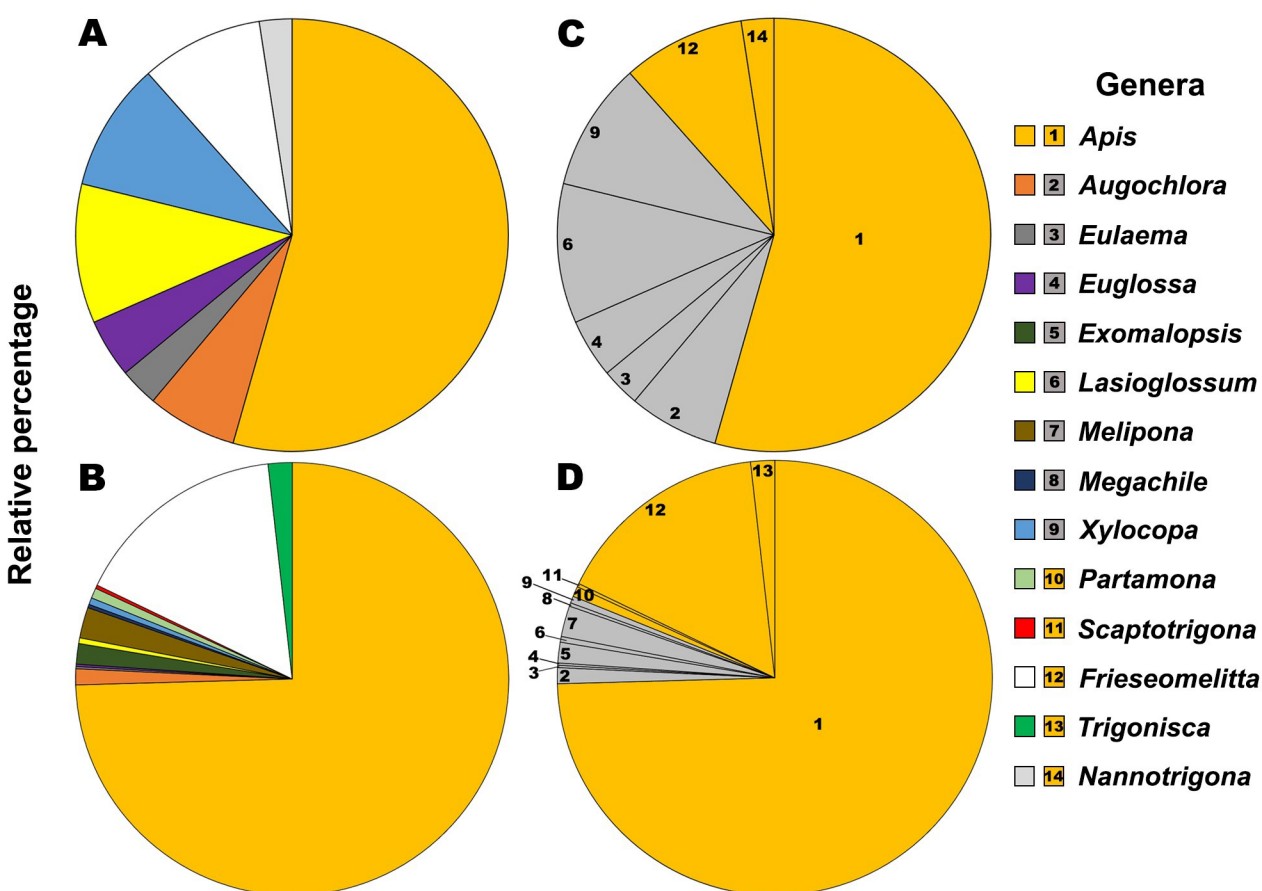

**Fig 1.** Pie charts showing the relative proportion of bee genera (in different color) visiting eggplant (A) and annatto (B) flowers in Yucatán, Mexico. In pies (C) eggplant and (D) annatto, the genera are classified as sonicating (in grey) and non-sonicating (in orange), to represent the relative proportions of both bee types in each crop. Numbers on pies indicate the different genera.

group, as they were not present in sufficient number to conduct experiments of controlled access to flowers.

**Annatto.** We used a similar method to the one used in eggplant to assess bee diversity and abundance in annatto. During a 30 min walk, 40 plants were visited by one trained observer to record the different types of bee present on the flowers. The surveys were conducted on eight separate days, every two hours between 08:00 and 14:00 h, which covers the period of more abundant annatto flowers visitors in Yucatan [5,33]. The time spent in data registration was compensated too, so that an effective 30 min observation period was conducted per survey, meaning a total of two hours of daily observations and a total of 16 h recordings for this crop. Likewise, two video cameras were used to record the behavior of AHB and native bees when visiting the flowers, especially the mechanism for the removal of pollen, resulting in a total of 16 h of recordings (n = 64).

Similar to the situation on eggplant, a first plot of the average daily abundance of specimens per genera showed that AHB were the dominant annatto floral visitors with 75% of all recorded visits (Fig 1B). Nevertheless, in annatto, NBZ represented only 6% compared with 18.9% of non-sonicating native bees (NB) (Fig 1D). As in annatto NBZ were not present in sufficient numbers, we were not able to conduct experiments of controlled access to flowers comparing AHB with NBZ as we did in eggplant, and we could only compare the AHB with native bees in general (NB).

In a further analysis, we compared the frequency of specimens in the three functional groups (AHB, NB and NBZ) on eggplant and annatto flowers across the day. To compare the frequency of bees per functional group in each crop, we used an ANOVA with two factors (bee group and hour) within blocks (day) followed by Tukey multiple comparison tests. The statistical model used was: $Y_{ijk} = \mu + \tau_i + \beta_j + (\tau\beta)_{ij} + \delta_k + \varepsilon_{ijk}$, where $\tau_i$, $\beta_j$ and $(\tau\beta)_{ij}$ represent the effect of bee group, hour and their interaction, respectively, and $\delta_k$ the effect of day. The frequency of visits to annatto was transformed using the natural logarithm ln(frec+1), due to the absence of specimens of some bee groups at some hours.

For the tests we used the statistical package Statgraphics Centurion 19 [36].

## Fruit and seed production by AHB and native bees

**Eggplant.**   To assess the pollination effect of NBZ and AHB we conducted experiments controlling the access of both functional groups to the crop flowers. For this, we randomly selected 15 eggplants and used bags made of cotton cloth (10 x 10 cm) to exclude floral buds close to open (anthesis). On the day of anthesis, the experimental virgin flowers were exposed to allow single visits by either NBZ or AHB. After each experimental flower received a single visit by a bee of one of the two functional groups, it was bagged again to restrain the access of other floral visitors. As controls, three flowers in each plant remained bagged (acronym EXC) and the same number were permanently exposed to open pollination (OPEN). To assess the effect of increasing number visits by AHB on pollination, we exposed virgin flowers of the experimental plants to visits of AHB only. For this, we exposed individual virgin flowers and awaited until they received one, two or three visits by AHB alone (treatments AHB1, AHB2 and AHB3, respectively). After being visited by the specific number of AHB, the experimental flowers were excluded again. If during the exposure to AHB visitation any other visitor contacted the flower, it was discarded. We did not include a treatment with multiple NBZ visits because their frequency was low and it was difficult to have NBZ visits only in the presence of AHB. To estimate the overall pollination service to the crop we included an additional treatment that consisted in surplus hand-pollination of permanently open flowers (PLUS) in all experimental plants. For this, we collected the pollen from 100 flowers, from 25 plants (different to the ones used in the bagged experiments) by means of an electric toothbrush. The collected pollen was pooled and was applied to the stigma of three recently opened flowers on each of the experimental plants by means of a fine brush. These flowers remained exposed during the whole experiment to receive unrestricted visits in addition to the artificial pollination provided. At the end of the experiments, we had three flowers from each of the seven treatments present *per* plant and a total of 45 flowers *per* treatment.

Eight weeks after the treatment was applied, we recorded the number of fruit produced per plant for each treatment, and estimated the percentage of fruit set per plant and treatment. The fruits were taken to the laboratory where they were weighed and the total number of seeds counted. As eggplant fruit can have numerous seeds [26], we used a proxy for the relative number of seeds. For this, we chose 20 fruits of different sizes, we weighed them and counted the exact number of seed in each. We then calculated the significance of the correlation between fruit weight and seed number using the Pearson's test and the value of Fisher's z to assess its significance. The correlation between weight and seed number was positive and highly significant ($r_{(18)} = 0.868$, $p = 0.001$; Fig 2). Given this significant result, we used eggplant fruit weight as an estimator of the pollination effect of the different treatments for the remaining experimental fruit.

**Annatto.**   We used a similar approach to eggplant to investigate the effect of AHB and native bees on the productivity of annatto. For this, we randomly selected 14 annatto plants

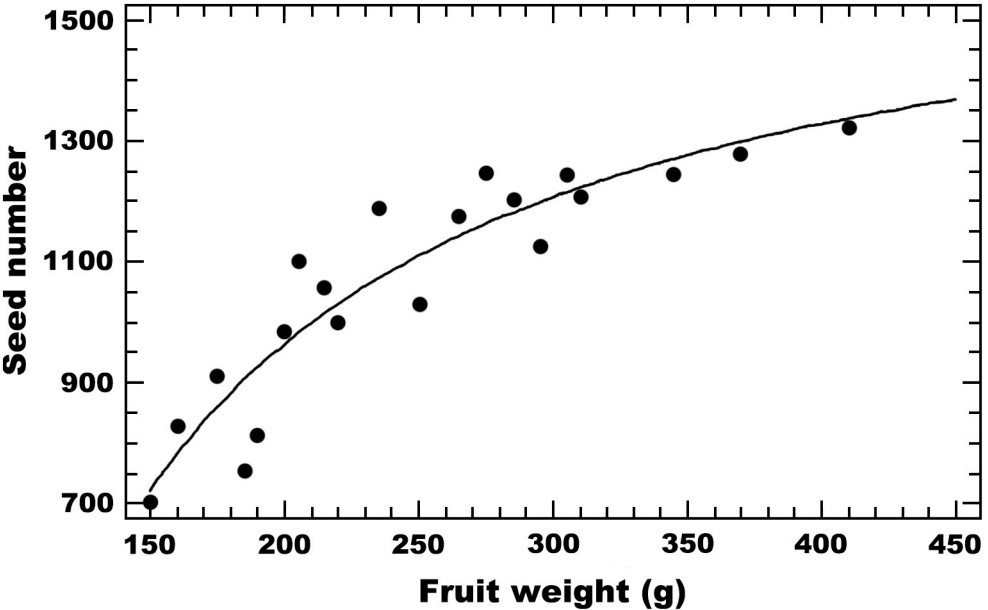

**Fig 2. Relation between fruit weight and seed number in eggplant fruit: Seed number = 1691.44 - (145373/ Fruit weight).**

and used cotton bags (15 x 15 cm) to exclude floral buds proximate to anthesis. In annatto it was not possible to evaluate the effect by NBZ due to their low abundance on this crop. Instead, we used visits in general by native bees, either buzz-pollinating or not (NB) to compare with AHB. To measure the effect of increasing number of AHB visits, we also exposed annatto virgin flowers to different visit number of AHB (AHB2, AHB3 and also AHB4) and used control flowers under the treatments EXC and OPEN. Likewise, we include the PLUS treatment to estimate overall pollination service to annatto following the same protocol used in eggplant to apply additional pollen to PLUS flowers. In annatto, we also had three flowers from each of, in this case, eight treatments *per* plant for a total of 42 flowers *per* treatment. We recorded the number of fruit produced and collected the ripe fruit eight weeks after. The number of seed per harvested fruit was counted in the lab.

The data for pollination treatments did not meet the criteria for normality in both, eggplant and annatto. To evaluate the hypothesis that all treatments produced similar weight of fruit in eggplant and seed in annatto, the data were compared by means of a Friedman test followed by a multiple comparison test. The Friedman test is a non-parametric statistical test similar to the parametric repeated measures ANOVA applicable to block designs [37]. Plants represented the blocks including each of the seven and eight treatments used in eggplant and annatto, respectively. The significance level was adjusted for each comparison using the Bonferroni method. For the comparison we used the statistical package Statgraphics Centurion 19 [36]. We plotted the results of fruit set and weight of the different pollination treatments using Box-plotR [38].

For a raw estimation of the contribution of different bee gilds on crop productivity, in eggplant we multiplied the mean number of fruit produced per gild treatment across plants (representing the fruit produced per each three flowers) by the average weight of those fruit. Then we calculated the percentage of difference of each of the treatments with respect to the PLUS treatment, which represents the maximum productivity that could be attained for this crop. Similarly, in the case of annatto we multiplied the mean number of seed produced per plant by

the number of fruit produced per treatment per plant. We then estimated the difference with the PLUS treatment.

## Pollination service

As a general measure of efficiency, in both, eggplant and annatto, we calculated Spears [39] single-visit pollination efficiency index (*PE*).

To obtain the seed number of the experimental eggplant fruit we used a best regression model of seed number *vs.* fruit weight, as:

$$\text{Seed} = 1691.44 - \left( \frac{145373}{\text{Fruit weight}} \right)$$

This final model was selected after comparing 27 models, in accordance to the highest value of its coefficient of determination ($R^2$ = 85.82%) and distance to the straight-line model ($R^2$ = 76.45%), that approximates the relationship between the dependent and the independent variables.

The index of pollinator efficiency was calculated with the Spear's [38] formula, *PE* = (*Pi-Z*)/(*U-Z*), where: *Pi* is the average seed number of fruit produced from flowers that received one visit by pollinator *i*, *Z* is the average seed number produced from flowers that received no visits and *U* is the average seed produced from flowers that received unrestricted visits. We also estimated the efficiency of successive visits by AHBs to eggplant and annatto flowers.

To estimate the overall pollination service provision (PSP) for the crops, we used the modified formula by Spears [39] in accordance to Landaverde et al. [4]. For this, we replaced the value of *Pi* and *U* in the original formula with the average seed of fruit produced from flowers that were left open and those that were hand pollinated, respectively.

## Results

### Frequency and behavior of floral visitors

**Eggplant.** There were significant differences in the number of visits to eggplant flowers between NBZ, NB and AHB (F = 338.17, *p* <0.0001), time of day (F = 48.52, *p* <0.0001) and the combination bee type time of day (F = 5.62, *p* = 0.0013; Fig 3A). The effect of block (day) was also significant (F = 10.24, *p* <0.0001). After Tukey multiple comparisons (Table 1), it was evident that AHB were significantly more abundant on eggplant flowers compared with NBZ and NB at all times during the day. The highest frequencies of individuals of the different bee types were recorded between 12:00 and 14:00 (Fig 3A and Table 1).

Visual material allowed identifying that during floral visits to eggplant, AHB landed on the anther cone and most of the time they used their mandibles to chew open the anther pore of eggplant flowers after which they used the glossa to extract the pollen. Less frequently, AHB used their front legs to glean pollen left upon petals or push the anthers by means of quick movements, as if drumming the anther cone (Fig 4A–4D and S1 Video).

**Annatto.** There were significant differences in the number of visits flowers among NBZ, NB and AHB (F = 115.97, p < 0.0001), time of day (F = 41.04, p < 0.0001) and the combination bee type by time of day (F = 2.27, p < 0.0450; Fig 3B). The effect of block (day) was also significant (F = 3.11, P < 0.0061). It was also evident that AHB were significantly more abundant on annatto flowers between 08:00 and 12:00 hours compared to NB and NBZ. At 14:00 hours there was a lower abundance of NBZ than that reported for AHB and NB, and the frequency of bees of these two groups did not differ at that time. The highest abundance in these functional groups were recorded between 10:00 and

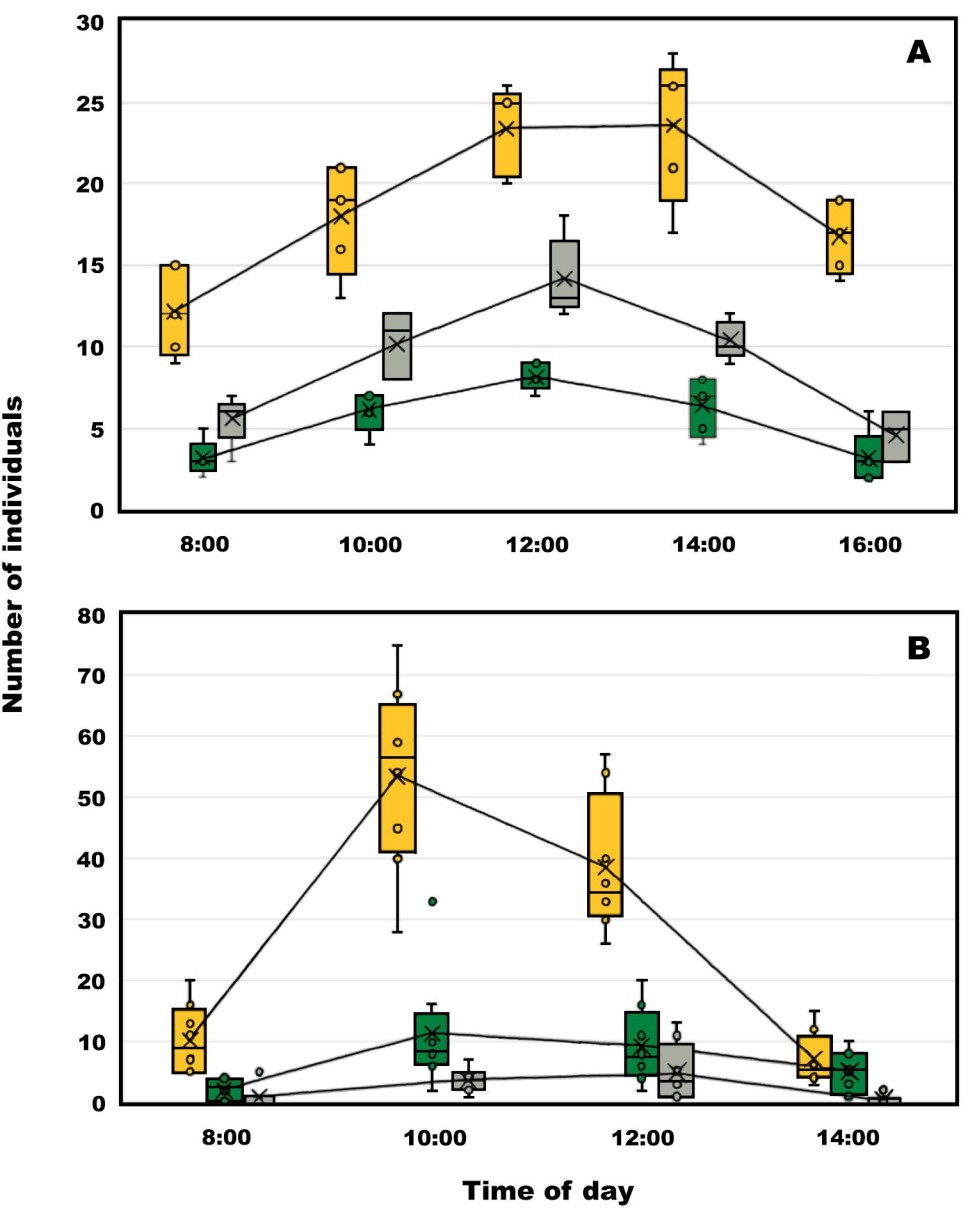

**Fig 3.** Distribution of visits of AHB (orange boxes), NB (green boxes) and NBZ (grey boxes) to eggplant (A) and annatto (B) flowers across different times of day in Yucatán, México. Lines inside the boxes represent the medians; box limits indicate the 25th and 75th percentiles; whiskers extend 1.5 times the interquartile range from the 25th and 75th percentiles.

12:00 hours (Fig 3B and Table 1). During floral visits to annatto, AHB landed on the petals and mostly used their front legs to glean pollen left upon the anther filaments. In contrast to eggplant visitation, they did not use their mandibles to chew open the anther pore of annatto flowers because it was apparently more difficult to reach and get a firm hold of them (Fig 4E–4H and S2 Video). In video it was possible to see how buzz-sonicating bees could release a cloud of pollen from annatto, but this was never seen during visits by AHB (S3 Video). It seems that AHB gleaned the pollen left after being released by sonicating bees.

**Table 1. Comparison of the average number of honey bees (AHB), native non-sonicating bees (NB) and native sonicating bees (NBZ) visiting flowers of eggplant and annatto in Yucatán, México.** Different letters within a time of day indicate significant differences at $p<0.05$.

| Bee type by time of day | Eggplant | Annatto |
|---|---|---|
| | Mean (S.E. = 0.842) | Mean (S.E. = 0.196) |
| AHB   8:00 | 12.2 a | 10.3 a |
| NBZ   8:00 | 5.6 b | 0.9 b |
| NB   8:00 | 3.2 b | 2.1 b |
| AHB   10:00 | 18.0 a | 53.5 a |
| NBZ   10:00 | 10.2 b | 3.8 b |
| NB   10:00 | 6.2 b | 11.4 b |
| AHB   12:00 | 23.4 a | 38.6 a |
| NBZ   12:00 | 14.2 b | 4.2 b |
| NB   12:00 | 8.2 b | 9.3 b |
| AHB   14:00 | 23.6 a | 7.1 a |
| NBZ   14:00 | 10.4 b | 0.4 a |
| NB   14:00 | 6.4 b | 5.1 b |
| AHB   16:00 | 16.8 a | |
| NBZ   16:00 | 4.6 b | |
| NB   16:00 | 3.2 b | |

## Fruit and seed production by AHB and native bees

**Eggplant.**   The numbers of fruit produced was statistically different among treatments ($F_r$ = 65.789, d.f. = 6, $p$ <0.001). The lowest number of fruits was produced in treatments EXC and AHB1 (Fig 5A and Table 2). Eggplant produced fruit even when flowers were excluded (EXC), but the production was low with only 17. 6% of flowers setting fruit. A similar result was obtained by AHB1 in single visits with 22.3% of fruit set. However, the number of fruits increased significantly but did not differ among treatments PLUS, OPEN, NBZ and AHB3. Interestingly, the number of fruit set by AHB3 was not different to that produced by NBZ on single visits and from the two treatments including permanently exposed flowers. Notably, the difference in fruit produced per plant by EXC and AHB1 compared with the other treatments was ca. four-fold.

Interestingly, there were significant differences among treatments when comparing the weight of the fruit ($F_r$ = 84.481, d.f. = 6, $p$ <0.001). PLUS and OPEN produced the heaviest fruit and were not statistically different from each other. The fruits with the lowest weight were produced by EXC and AHB1 (Fig 6A and Table 2), which ca. 50% difference in weight compared with PLUS. Noteworthy, AHB3 produced fruit of similar weight to that of NBZ and OPEN, but the average was significantly lower than PLUS.

**Annatto.**   The numbers of fruit produced was statistically different among treatments (Fr = 16.231, d.f. = 7, $p$ < 0.0231). The lowest number of fruits was produced by EXC. Notably, individual and multiple visits by AHB resulted in similar numbers of fruit set and did not differ from individual visits by NB as well as OPEN (Fig 5B and Table 3). It is important noting that at the time of collection of the ripe fruit at eight weeks, the PLUS treatment had the highest fruit set compared to other treatments. Notably, the number of seeds per ripe fruit showed significant differences among treatments (H = 47.9659, d.f. = 7, $p$ <0.0001), being higher in PLUS and OPEN compared to the other six treatments (NB, AHB1, AHB2, AHB3 and AHB4), which did not differ among them (Fig 6B and Table 3).

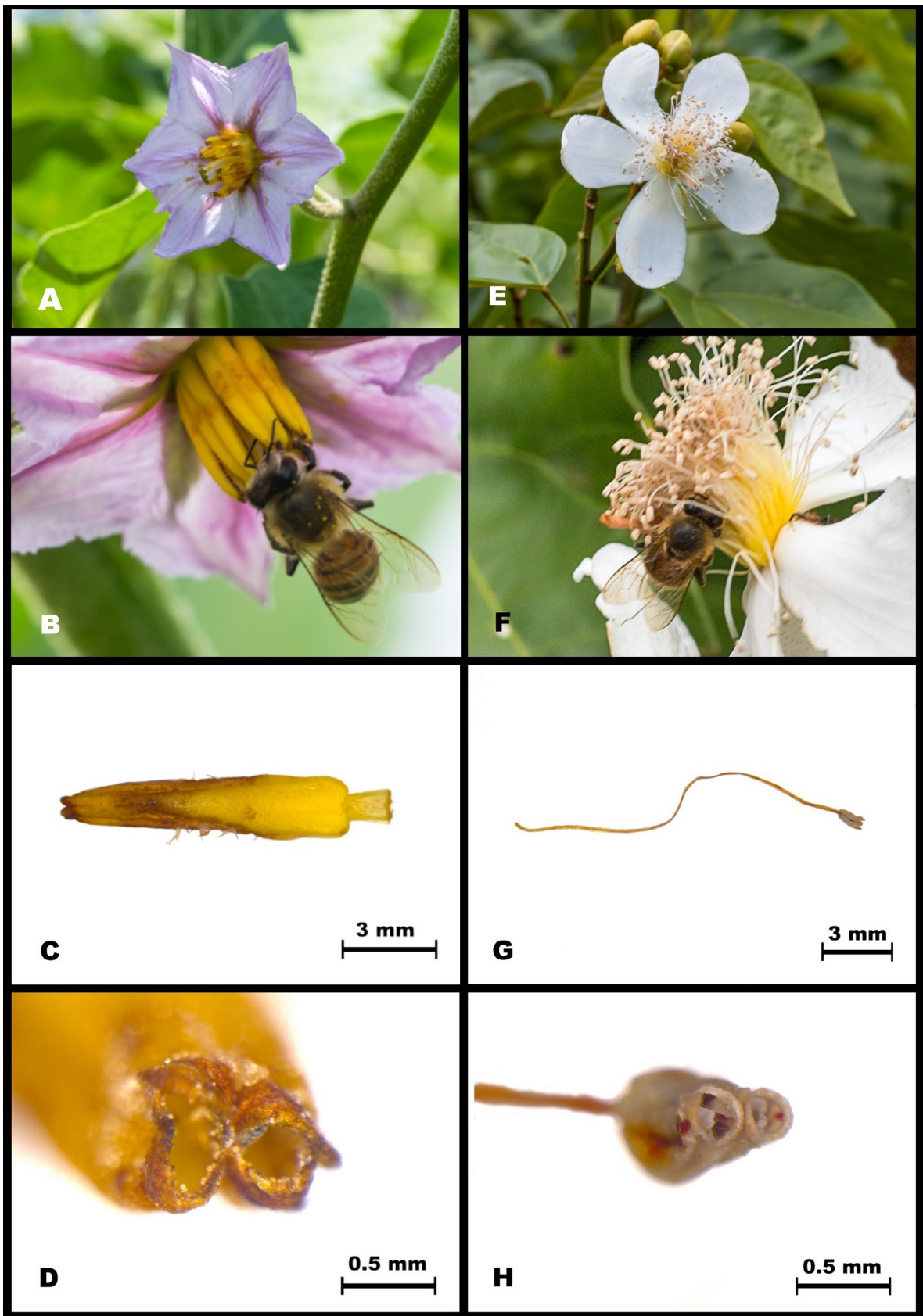

**Fig 4.** The flower of eggplant (A) and the behavior of AHB *Apis mellifera* during floral visit (B). Cone anther of eggplant flower (C) and its poricidal dehiscence (D). The comparison with annatto flower (E) and AHB visit (F). The structure of an annatto anther with elongated filament is shown on (G) and pore of the theca (H).

## Pollination service

**Eggplant.** The average seed number estimated for the experimental fruit (see previous section) was used to calculate Spear's pollination efficiency' index for single-visit performed by NBZ and AHB and multiple visits of the latter. On single visits the efficiency of AHB1 was 0.24, compared with NBZ that was 0.63. Nonetheless, multiple visits by AHB2 and AHB3

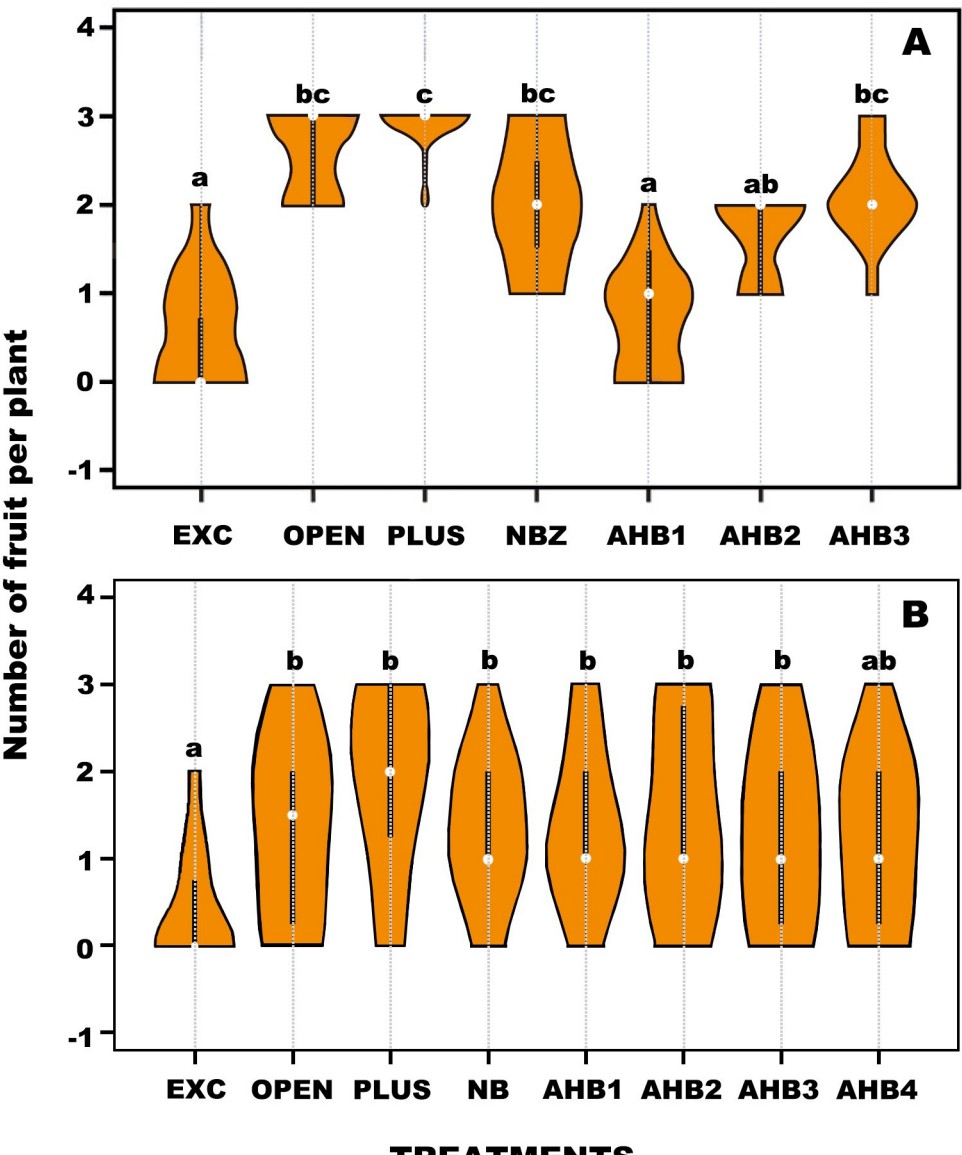

**Fig 5.** Violin plots of number of fruits produced in A) eggplant and B) annatto after the application of treatments for controlled flower access (see text for acronyms). White circles show the medians; box limits indicate the 25th and 75th percentiles. Different letters indicate significant differences among treatments at *p*<0.01.

**Table 2. Comparison of the average number of fruit and their weight obtained in seven treatments (see text for acronyms) applied to eggplant flowers.** Estimations of Pollination Efficiency Index at single visits by NBZ and individual and multiple visits by AHB are also presented, together with estimation of overall Pollination Service Provision (PSP).

| Treatment | Number Fruit (three flowers per plant) (NF) | Fruit set (%) | Fruit weight (g) (FW) | Estimated productivity (g) per plant (NF x FW) | Average Seed number | Spear's Efficiency Index | PSP |
|---|---|---|---|---|---|---|---|
| EXC | 0.53 (0.17) a | 17.6 | 203.2 (3.39) a | 107 (7.7%) | 973.52 | | |
| OPEN | 2.60 (0.13) bc | 86.6 | 389.4 (2.82) cd | 1012.44 (72.8%) | 1316.82 | | 0.83 |
| PLUS | 2.93 (0.07) c | 97.6 | 474.3 (2.65) d | 1389.69 (100%) | 1384.81 | | |
| NBZ | 2.00 (0.20) bc | 66.7 | 278.2 (5.21) bc | 556.4 (40%) | 1192.59 | 0.63 | |
| AHB1 | 0.67 (0.16) a | 22.3 | 231.7 (3.71) a | 155.23 (11.2%) | 1058.55 | 0.24 | |
| AHB2 | 1.67 (0.13) ab | 55.7 | 253.5 (2.65) ab | 423.34 (30.5%) | 1118.22 | 0.42 | |
| AHB3 | 2.13 (0.13) bc | 71.1 | 273.6 (6.33) bc | 582.76 (41.9%) | 1155.03 | 0.52 | |

Different letters within a column indicate significant differences at p<0.01. The numbers in brackets indicate the standard error, except for Estimated Productivity, where it indicates the relative % of each treatment in relation to the PLUS treatment that represents the theoretical maximum.

resulted an increasing rate of efficiency (0.42 and 0.52, respectively). Finally, we estimated the Pollination Service Provision for the crop, which was 0.83. This value reflects no substantial pollination deficit. However, it is important to note that the estimated productivity for the crop, would be ca. 27% less in relation to a maximum (the difference of fruit weight between PLUS and OPEN treatments) (Table 2).

**Annatto.** The seed production efficiency of the single visit by NB (0.46) was similar to the single visit to AHB1 (0.45). The single visits of NB and AHB1 did not differ to the multiple visits of AHB2 (0.48) and AHB3 (0.49), with a slight increase in AHB4 (0.54). Apparently, multiple visits by AHB did not result in an increasing rate of efficiency in annatto. The PSP in annatto was 0.615, which reflects a pollination deficit. Noteworthy, the difference in estimated productivity between OPEN and PLUS treatments was ca. 47% in annatto confirming the deficit on pollination for this crop (Table 3).

## Discussion

Insufficient information is available on the pollination biology and pollinator requirements of many crops, especially when considering the joint contribution to pollination services by different pollinator species [25]. In our study, we evaluated the pollination services provided to buzz-pollinated eggplant and annatto by non-sonicating AHBs and native bees. Our goal was to assess the service of generalist honey bees to buzz-pollinated crops, which is of particular relevance to Latin America, an area with the largest density of honey bee colonies in the world [23,40,41].

Our analysis of bee frequency confirmed that AHB was the dominant floral visitor to eggplant and annatto, with more than half of the total visits registered on both crops. Noteworthy, in eggplant, NBZ were registered on flowers around 30% of the times, but this figure was considerably lower in annatto where they represented only 6% of the visits. Interestingly, different studies performed in Mexico have also shown that AHB are the dominant floral visitors of several buzz-pollinated crops, including tomato [13], and habanero pepper [4] in addition to annatto [5].

The predominance of AHB on buzz pollinated crops seems a result of the abundance of managed and wild colonies present in neotropical Mexico. The Yucatan Peninsula possess some of the largest densities of honey bee colonies in the world that can be up to 30/km$^2$

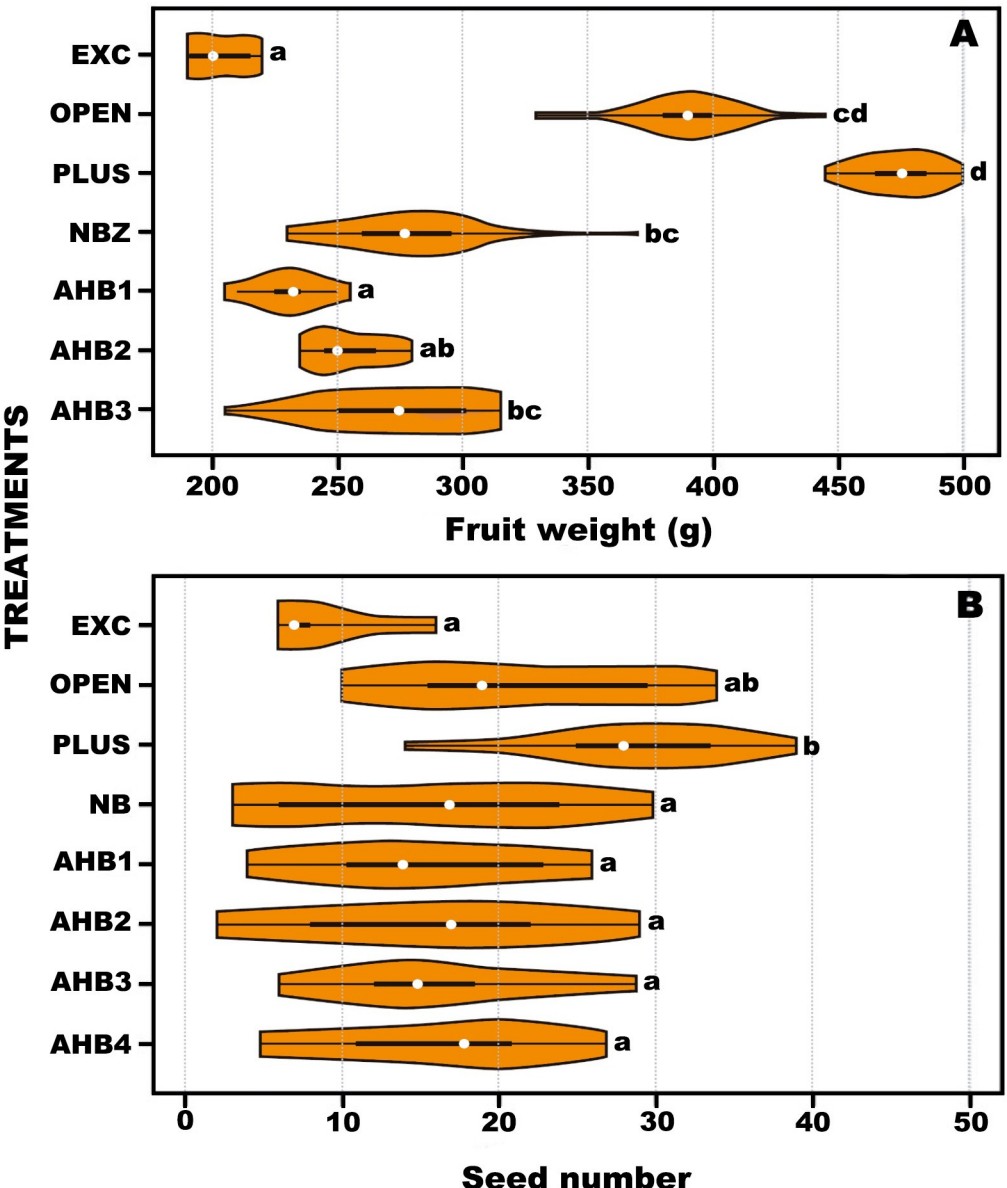

**Fig 6.** Violin plots comparing the weight of eggplant fruit (A) and seed number of annatto fruit (B) after the application treatments for controlled flower access (see text for acronyms). White circles show the medians; box limits indicate the 25th and 75th percentiles as determined by R software; whiskers extend 1.5 times the interquartile range from the 25th and 75th percentiles. Different letters indicate significant differences among treatments at $p < 0.01$.

[16,42]. In their search for food, generalist AHB expand the diversity of visited floral resources, thus, becoming predominant on various crops [5,43].

Regarding the efficiency of AHB, we confirmed that on single visits to flowers of eggplant and annatto [5], they were poorly efficient, being similar to the production of fruit and seed obtained in excluded treatments with no access of pollinators. However, we additionally assessed the contribution of AHB after multiple floral visits, providing a more realistic image of the service they may supply [44]. Notably, in eggplant, although AHB were inefficient on individual visits to flowers, when the number of visits per flower increased, a notorious additive effect was observed, resulting in a significant increase of fruit weight after three visits,

**Table 3. Comparison of the average number of fruit and their seed number obtained in eight treatments (see text for acronyms) applied to annatto flowers.** Estimations of Pollination Efficiency Index at single visits by NB and individual and multiple visits by AHB are also presented, together with estimation of overall Pollination Service Provision (PSP).

| Treatment | Number Fruit/plant (NF) | Fruit set (%) | Average seed number (SN) | Estimated productivity (NF x SN) | Spear's Efficiency Index | PSP |
|---|---|---|---|---|---|---|
| **EXC** | 0.36 (0.17) a | 11.9 | 10.20 (1.83) a | 3.64 (6.5%) | | |
| **OPEN** | 1.36 (0.29) b | 45.2 | 21.68 (1.76) ab | 29.42 (52.9%) | | 0.615 |
| **PLUS** | 1.93 (0.29) b | 64.3 | 28.85 (1.16) b | 55.64 (100%) | | |
| **NB** | 1.36 (0.23) b | 45.2 | 15.47 (1.86) a | 21 (37.7%) | 0.46 | |
| **AHB1** | 1.36 (0.25) b | 45.2 | 15.32 (1.66) a | 20.79 (37.4%) | 0.45 | |
| **AHB2** | 1.5 (0.31) b | 50 | 15.67 (1.86) a | 23.51 (42.2%) | 0.48 | |
| **AHB3** | 1.29 (0.29) b | 42.9 | 15.78 (1.47) a | 20.29 (36.5%) | 0.49 | |
| **AHB4** | 1.21 (0.26) ab | 40.5 | 16.35 (1.69) a | 19.85 (35.7%) | 0.54 | |

Different letters within a column indicate significant differences at p<0.01. The numbers in brackets indicate the standard error, except for Estimated Productivity where it indicates the relative % of each treatment in relation to the PLUS treatment that represents the theoretical maximum.

similar to that of NBZ on single visits. It is interesting noting that a former investigation in greenhouses showed a similar production comparing honey bees with sonicating bumble bees [45]. Thus, in open fields frequent visits by honey bees could compensate their poor individual efficiency and produce fruit of good weight after repeated visits to the same flower.

Notably, in contrast to the results on eggplant, the increasing number of AHB visits to flowers did not have an additive effect on annatto seed production. Seed produced in annatto after multiple visits by AHBs was not significantly different from flowers that were excluded from pollinator access. As a result, the overall pollination service and productivity estimated to annatto was comparatively lower than to eggplant.

The different effects of AHB on eggplant and annatto could have explanation on the possibility to access pollen by non-sonicating bee species, because of the contrasting structure of the anther of both flower types. In the case of eggplant, the flowers have their anthers fused forming a cone around the pistil. On this type of *Solanum* flowers, anthers open through apical pores, and poricidal-longitudinally dehiscence [46]. On *Solanum* flowers with anther cones, honey bees land on the cone and can easily access the tip [45,47,48]. We observed that AHB used their mandibles to chew open the anther pore of eggplant flowers and then use the glossa to extract the pollen contained. Such a behavior, called "milking" of the anther [49], allows honey bee access to concealed pollen from this type of anther. AHB also used their front legs to glean pollen left upon petals. Less frequently, we saw AHB using the first two pair of legs to push the anthers by means of quick movements, what is defined as "striking" or "drumming" of the anther [49,50]. In addition, the anther cone structure of *Solanum* flowers increases pollen release, whereas this is more difficult in species with a free anther architecture [51]. Evidently, honey bees are capable of removing pollen from *S. melongena* and they can pollinate them. Similar results were obtained in Brazil with the stingless bee *Paratrigona*, which in spite of not having the ability to sonicate, was an effective pollinator of eggplant [52].

In contrast to eggplant, the anthers of annatto flowers are not fused. The horse shoe-like theca is found at the end of a simple uncombined filament [53]. Notably, *Bixa* anthers have a restricted slit dehiscence that seem to require exclusive sonication for pollen release [54]. Due to such a particular structure of the anther and the fact that it is found at the extreme of an elongated filament, honey bees cannot easily access them and thus, cannot milk annatto anthers as they do with eggplant anther cones. Instead, we observed that AHB seem only capable of collecting pollen from annatto flowers already released in previous visits by sonicating bees.

The inability of honey bees to efficiently pollinate buzz-pollinated plants on individual visits has been evident in crops like *S. lycopersicon* [9], and *Vaccinium* berries [55,56]. Notably, in open cultivated *Vaccinium*, it was recently reported that, similar to our results, multiple visits by honey bees to flowers also have an additive effect, even comparatively similar to that of buzz-pollinating bumble bees [56]. We found a similar trend in eggplant, although AHB were not efficient on single visits, the quantity and quality of fruit increased with the number of visits. Thus, the abundance of honey bees seems to compensate individual low efficiency [47,55–57]. This finding is important in the light of decreasing pollinators worldwide [58] and the abundance of honey bees in the neotropics [23]. For some economically important buzz-pollinated crops, commercial and wild honey bees could grant resilient pollination, an important step towards increasing sustainable agriculture production [3,59]. Nonetheless, the potential negative effect of overwhelming honey bee visits to buzz-pollinated flowers remains to be investigated [60]. For instance, frequent visits by non-buzzing *Trigona* bees can damage buzz-pollinated flowers and compromise pollination by reducing the visual attractiveness of these flowers to other more effective pollinators [61].

In contrast with the positive effect on eggplant, we found that AHB do not compensate pollination service with multiple visits on annatto. As AHB seem to mainly obtain pollen from flowers already released by previous visitors, it would be crucial to preserve native buzz pollinating bees for sufficient pollination service to this crop. Indeed, when buzz-pollinating species (like *Melipona*) had been present or introduced to annatto, honey bees could provide a complementary service, gleaning the pollen released by buzz-pollinating bees and distributing it to other flowers [5]. However, when NBZ are absent or at low numbers, honey bees may not be capable of obtaining annatto pollen by themselves. This fact could make the production of annatto more dependent on the presence of NBZ compared with crops with anther cones, such as eggplant. Thus, it would be crucial to evaluate if the overwhelming presence of AHB can affect the presence or activity of NBZ through exploitative competition in buzz-pollinated crops lacking anther cones [62,63].

Our estimations of PSP indicated no substantial pollination deficit in eggplant but a deficit in annatto. The difference in productivity between flowers that were hand pollinated and those that were left open confirmed that AHB dominance does not grant sufficient pollination. A rough estimate of productivity of our eggplant crop (in kg of fruit) suggested a 90% contribution of bees in general to the productivity of the crop (comparing the relative productivity of EXC to OPEN). In the case of the Yucatan Peninsula, 57% of that service could be provided by AHB (comparing the relative contribution of AHB3 to OPEN), which underlines their economic relevance as pollinators and the stability to some agricultural systems in the neotropics [3,17,27]. However, for annatto, the estimated PSP was comparatively lower. Notably, AHB visit contributed only an estimated ca. 36–42% to the productivity of annatto in the field (comparing the contribution of AHB1 and AHB4 to OPEN) and their recurrent visits did not substantially increase seed numbers. Thus, productivity in annatto, may be more dependent on preserving or promoting a richer and more varied community of bees.

In summary, our results indicate that honey bee numerical dominance is not equally beneficial to different buzz-pollinating crops in the neotropics [6,56]. We suggest that differences in

the structure of the anther can influence the possibility of pollen removal by honey bees (and possibly other non-sonicating species), thus, resulting in differences on the pollination services that they can provide. It seems that honey bees could help to ensure pollination stability on some crops that require specialized pollination, but this would strongly depend on their particular floral structure.

It is generally believed that increased pollinator diversity and abundance can improve pollination services [64]. The synergistic combination of *A. mellifera* and non-*Apis* bees, represents in principle, a sustainable way to enhance crop pollination. Interactions among species can alter the behavior and their functional quality, which may result in better crop yield [41,59,65,66]. However, it is important to consider the potential negative effects of artificially increasing honey bee numbers in crops. As our results suggest, this practice may not be equally beneficial and the reproductive biology of particular crops should be pondered. In the neotropics, honey bee overdominance could also have not yet fully understood impacts on the native apifauna. There may be an optimum of honey-bee densities above which their effect may be detrimental, but it remains unassessed on most crops [22]. We suggest that to ensure resilient pollination services, the best option would be to promote the presence of a diversity of species on these and other crops [6,65,67]. Preserving natural biotopes around the cultivars to promote density and diversity of native bees, including specialized guilds, should be one essential aspect for a sustainable agriculture.

## Supporting information

**S1 Video. Behavior of Africanized *Apis mellifera* during floral visits to eggplant.** This behavior of chewing the tip of the fused anther cone to open the pores is commonly observed. Wondershare Filmora 11 video editor was used to stabilize the video.
(MP4)

**S2 Video. Behavior of Africanized *Apis mellifera* during floral visits to annatto.** The bees have difficulty to grab the anthers that could allow biting them and opening the pores. Wondershare Filmora 11 video editor was used to reduce to 0.25x the speed.
(MP4)

**S3 Video. Behavior of buzz-pollinating *Exomalopsis* bee during floral visits to annatto.** A cloud of pollen is visible during the floral visit as well as the characteristic buzzing sound. It seems the pollen left by sonicating bees is the only one that could be collected by *Apis mellifera*.
(MP4)

**S1 Data.**
(ZIP)

## Acknowledgments

Thanks to two anonymous reviewers for their insightful opinions and remarks on our ms. Our sincere thanks to Prof Robert J. Paxton (Halle) for suggestions on the experiments and to Dr David McFarlen (MSU) who provided valuable comments during draft writing.

## Author Contributions

**Conceptualization:** Franklin H. Rocha, José Javier G. Quezada-Euán.

**Data curation:** Franklin H. Rocha, Daniel N. Peraza.

**Formal analysis:** Daniel N. Peraza, Salvador Medina.

**Funding acquisition:** José Javier G. Quezada-Euán.

**Investigation:** Franklin H. Rocha, Daniel N. Peraza, José Javier G. Quezada-Euán.

**Methodology:** Franklin H. Rocha, Salvador Medina, José Javier G. Quezada-Euán.

**Software:** Salvador Medina.

**Supervision:** José Javier G. Quezada-Euán.

**Visualization:** Daniel N. Peraza.

**Writing – original draft:** Franklin H. Rocha, José Javier G. Quezada-Euán.

**Writing – review & editing:** Franklin H. Rocha, Daniel N. Peraza, Salvador Medina, José Javier G. Quezada-Euán.

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
