## [Decision Letter · Decision Letter 0]

6 Nov 2022

PONE-D-22-23499Pollination service provided by honey bees to buzz-pollinated crops in the NeotropicsPLOS ONE

Dear Dr. Quezada-Euan,

Thank you for submitting your manuscript to PLOS ONE. After careful consideration, we feel that it has merit but does not fully meet PLOS ONE’s publication criteria as it currently stands. Therefore, we invite you to submit a revised version of the manuscript that addresses the points raised during the review process. Both reviewers list a number of points that need to be improved, and particularly any problems of discrepancy between the raw data and summary data or non-transparent raw data must be fixed. 

We look forward to receiving your revised manuscript.

Kind regards,

Olav Rueppell

Academic Editor

PLOS ONE

Journal Requirements:

To project CONACyT-SADER 291333 “Manejo sustentable de polinizadores”, for supporting this research and CONACyT study grants to FHR and DNP.

6. Please include your tables as part of your main manuscript and remove the individual files. Please note that supplementary tables (should remain/ be uploaded) as separate "supporting information" files

Reviewers' comments:

Reviewer's Responses to Questions

**Comments to the Author**

1. Is the manuscript technically sound, and do the data support the conclusions?

Reviewer #1: Yes

Reviewer #2: Partly

2. Has the statistical analysis been performed appropriately and rigorously? 

Reviewer #1: I Don't Know

Reviewer #2: Yes

3. Have the authors made all data underlying the findings in their manuscript fully available?

Reviewer #1: No

Reviewer #2: No

4. Is the manuscript presented in an intelligible fashion and written in standard English?

Reviewer #1: Yes

Reviewer #2: Yes

5. Review Comments to the Author

Reviewer #1: This is a very interesting study investigating the contribution of native and introduced bees on the pollination of two crops associated with buzz pollination. The study is extremely valuable in its contribution to the pollinator visitation, and crop yield consequences of visits by different pollinator guilds and will make a very valuable contribution to the literature. Below I provide some major and minor suggestions.

Major suggestions

1) Data. The data provided in the Supplemental Information is not of sufficient quality to allow replicating the analyses of the paper. You must include all necessary data, with a clear explanation of column names, units and how they were calculated. The objective of providing the data for your study should be to allow others to replicate your analysis and enable subsequent investigations, and now the Excel file is poorly organised and described. Maybe the necessary information is there, but without a README file is hard to tell what is what. https://journals.plos.org/plosone/s/data-availability

2) Introduction. Please provide a brief overview of native bee diversity in the study region. This will be very useful when trying to understand what level of diversity is encompassed in the term you use (native bees) throughout the rest of the paper. The honeybee vs. native bee seems a simplistic comparison and you need to give more context here.

3) Details of methods, sample sizes and analyses are missing in some parts. See comments below for specific suggestions.

Minor suggestions

4) Abstract. Consider removing the acronyms in the abstract.

5) Line 46. “Few studies…” Perhaps, but it might be a good idea to add other references as a single citation here may give a bias idea of the numerous studies of pollination services in the tropics.

6) Line 60. Here you should mention that honeybees are introduced in the Americas. Is not obvious to all readers.

7) Line 64. What do you mean by “natural” abundance?

8) Suggested change: “The Solanaceae is…”

9) Line 78. Introduce authority and year for eggplant as you do for achiote in line 85

10) Line 79. Remove “thus” as self-compatibility does not imply self-pollination

11) Line 89. Which region is the major producer of achiote?

12) Line 94. AHB. I am not a fan of acronyms. Could you simply define and use “honeybees” throughout the paper?

13) Line 94. No question has been formulated. Rephrase.

14) Line 116-117. More details needed. How many observers/cameras? Provide a standardised unit for observations (person-hours, etc.).

15) Line 117. How was it determined if a bee buzz pollinated. Details needed as this is essential for the functional classification done in the next paragraph.

16) Line 119. Why was this initital analysis done? What was the sample size? It may be best to formally analyse this and report in the results with sample size details, etc.

17) Lines 121-122. How were these determinations done? Are voucher specimens collected and stored? How many species per genus? The information on determination appears briefly later. Consider re-organising.

18) Line 147. Was time nested in day? More details of statistical model are needed. Consider providing the full model used, and make sure it matches the results (eg. lines 233-236). You need to also explain whether the model assumptions are met.

19) Line 152. “Both functional groups” I think you mean the three functional groups? Clarify.

20) Line 162. Why 2-3 visits? Was visit number included in the statistical model, and if not why not?

21) Line 193. Not sure what the AHB2, AHB3,… mean.

22) Line 198-200. Explain what the Friedman test is.

23) Line 34-347. Please rephrase. Most genera in Solanaceae are not buzz pollinated. Within Solanum, not all species have anthers fused into a cone (the majority do not).

24) Line 414. In this sentence, consider including in the discussion a more nuanced evaluation of the benefit of honeybees for “sustainable” pollination (as you state at the end of the paragraph). To the extent that honeybees impact native bees, it might not be very sustainable in a broader context.

Reviewer #2: The study provides some very interesting information about how non-native, feral honey bee colonies affect buzz-pollinated crops. Your analyses support your conclusions that 1) honey bees have become the dominant visitors to these crops, 2) that despite their lack of buzz pollination behavior honey bees seem to pollinate eggplant effectively by visiting multiple times, and 3) that honey bees may be decreasing pollination success for annato by competing with more effective native pollinators. These conclusions contribute interesting and useful information for anyone interested in agriculture or bee conservation in the tropics, and should be published. It was especially interesting to see how multiple visits by honey bees seem to compensate for their inability to vibrate the anthers of eggplant flowers but there is no similar effect in annato flowers. The experiments allowing 1, 2, or 3 honey bee visits and counting or estimating seed production are impressive. I imagine that took a lot of careful work.

My main suggestion would be to double-check your bee abundance data for the 3 guilds and to make sure that you provide all data for figure 5. Table 1 does not match the data in your eggplant and annato data files. I have some suggestions below to add or clarify information at particular lines and figure/table captions.

You do not need to do this, but I would highly recommend providing your R code and providing your data in separate (ideally csv) files that can be read directly from your R code. This would help make your study more reproducible and easier to interpret. It would also help you to double-check your results (or clarify what data you included in each summary/analysis).

Introduction-

Good introduction, sets up the importance/questions well

Line 44-45: I know you understand that many crops are self-compatible or wind pollinated, but the way you say this, it makes it sound like all crops need visits from pollinators.

Materials and Methods-

Overall thorough description of your methods, but there are a few places that need clarification

Lines 116-117: Did you pause walking the transects each time you saw a bee and record it?

Line 154: When I hear the word proximate, I usually think of close in spatial location, but here you mean close in time, correct?

Line 198: So the data violated the assumption of normality and you used a non-parametric test instead?

Line 218: I'm assuming the dotted line is this function? It looks like a good fit, but it would be helpful to include a little more detail about how you decided that this was the best fit

Lines 219-228: It was helpful that you included a description of the formula here

Results-

Overall thorough and clear, but there seems to be a problem in one of the tables and the captions need more detail

Line 296: So am I correct that your point here is that even though the increase in seed production with unlimited access to the pollinators was statistically indistinguishable from the seed production produced by hand pollination, it's still possible that a different community of pollinators (maybe one with more native bee species) could have done a better job of providing pollination services to the eggplants?

Line 303: Do you mean 38% less than the maximum possible?

Table 1:

There seems to be a few errors in the eggplant column and many errors in the annato column when I compare them to your data files. Please double-check these!

What does EE refer to?

Picky comment- I would recommend presenting times of day in the format 8:00 or 16:00 rather than 8h and 16h because at first I thought those were durations

Figure 1:

This figure contains interesting information about the diversity of pollinators you found, but I wonder if it wouldn't be better to present pie charts because it would better convey the relative proportions. But that's personal preference.

Figure 2:

It would help to put in the caption what the dotted line represents (maybe even including the equation you present in the text) because saying correlation makes me expect a straight line here (clearly you didn't just look for a linear correlation, but it would help to clarify that).

Figure 3:

This figure is very clear and contains a lot of information about the variation and trends

Figure 4:

This figure is very helpful, providing context for the methods and discussion. Very nice photos of the anthers!

Figure 5:

Why are all the y values between 0 and 3? Was that the range for both species or did you normalize somehow? I don't see where those numbers are in your eggplant data file.

Table 2:

The percentages of the hand-pollinated results are very helpful. I think the letters to indicate significant differences would be good to add to the figure.

Table 3:

It would helpful to switch the NB and PLUS rows so they match the order in the figure

S1 Video:

Very interesting behaviors! You don't need to do this, but I wonder if it would make the video more likely to be used by others to either slow the it down or use video stabilization so it is easier to focus on the bee's behavior rather the flower moving in the wind (I know ffmpeg has a way to do this, although I haven't used it: https://www.paulirish.com/2021/video-stabilization-with-ffmpeg-and-vidstab/). If you do either of those things, please include a description of what you did in the video caption

S2 Video:

This looks like it was slowed down. It would be helpful to indicate how you did that (what % change, when in video)

S3 Video:

The cloud of pollen is quite striking. I'm not sure that you need the arrow, but if you keep it, I think it would be good to say in the caption that it is pointing to the cloud of pollen

Discussion-

It was helpful that you related your findings to previous literature on the pollination of buzz-pollinated crops and talked about potential reasons for the difference in efficiency of honey bees across the 2 crops. Your conclusions about supporting diverse native pollinators to maximize pollination services across crops seems reasonable.

Line 402-403: How are you calculating AHB contribution here? Is this because native non-sonicating bees were also poor pollinators per trip?

6. PLOS authors have the option to publish the peer review history of their article (what does this mean?). If published, this will include your full peer review and any attached files.

Reviewer #1: No

Reviewer #2: No

---

## [Author Response · Author response to Decision Letter 0]

20 Dec 2022

WE WOULD LIKE TO THANK THE EDITOR AND THE TWO REFEREES FOR THEIR THOROUGH REVIEW AND FEEDBACK ON OUR MS ENTITLED “Pollination service provided by honey bees to buzz-pollinated crops in the Neotropics”. IN THE FOLLOWING PARAGRAPHS WE ADDRESS EACH COMMENT AND SUGGESTION RAISED BY THE EDITOR AND BOTH REVIEWERS.

RESPONSES TO EDITOR’S COMMENTS

R= We have consulted the style templates and have made the pertinent style corrections to meet PLOS ONE’S requirements.

2. In your Methods section, please provide additional information regarding the permits you obtained for the work. Please ensure you have included the full name of the authority that approved the field site access and, if no permits were required, a brief statement explaining why. R= A brief explanation for access to field locations for which no permit was required has now been included in the Methods section.

R= We have corrected the funding information, removed it from the Acknowledgements section and provided the correct grant numbers for the awards we received in the ‘Funding Information’ section. 

To project CONACyT-SADER 291333 “Manejo sustentable de polinizadores”, for supporting this research and CONACyT study grants to FHR and DNP. 

R= We have corrected the funding information removing it from the Acknowledgements section and provide the correct grant numbers for the awards we received in this cover letter

R= We have prepared the repository information to provide the respective accession DOI.

6. Please include your tables as part of your main manuscript and remove the individual files. Please note that supplementary tables (should remain/ be uploaded) as separate "supporting information" 

R= We have removed the files for tables and have included them in the main manuscript.

R= We have included captions for the Supporting information at the end of the ms as requested.

RESPONSES TO REVIEWERS

Reviewer #1: This is a very interesting study investigating the contribution of native and introduced bees on the pollination of two crops associated with buzz pollination. The study is extremely valuable in its contribution to the pollinator visitation, and crop yield consequences of visits by different pollinator guilds and will make a very valuable contribution to the literature. Below I provide some major and minor suggestions.

R= We are thankful for the time and insightful remarks and positive comment on our manuscript. We reply to each of your queries below.

Major suggestions

1) Data. The data provided in the Supplemental Information is not of sufficient quality to allow replicating the analyses of the paper. You must include all necessary data, with a clear explanation of column names, units and how they were calculated. The objective of providing the data for your study should be to allow others to replicate your analysis and enable subsequent investigations, and now the Excel file is poorly organised and described. Maybe the necessary information is there, but without a README file is hard to tell what is what. https://journals.plos.org/plosone/s/data-availability

R= We have updated the data files and better organized them including explanations of the data presented in each Excel book. We hope this will be satisfactory.

2) Introduction. Please provide a brief overview of native bee diversity in the study region. This will be very useful when trying to understand what level of diversity is encompassed in the term you use (native bees) throughout the rest of the paper. The honeybee vs. native bee seems a simplistic comparison and you need to give more context here. 

R=As requested by the reviewer, we have included information on the diversity of native bees in the Yucatan (L. 61-64).

3) Details of methods, sample sizes and analyses are missing in some parts. See comments below for specific suggestions.

R= We address these queries in accordance to each suggestion made by the reviewer

Minor suggestions

4) Abstract. Consider removing the acronyms in the abstract. 

R= we considered the reviewer suggestion but as we used many treatments, we decided to keep the acronyms for clarity of reading 

5) Line 46. “Few studies…” Perhaps, but it might be a good idea to add other references as a single citation here may give a bias idea of the numerous studies of pollination services in the tropics.

R= we added other references so not to give a biased idea as suggested by the referee, L. 48-49.

6) Line 60. Here you should mention that honeybees are introduced in the Americas. Is not obvious to all readers. 

R= thank you for the suggestion, the explanation has been included L. 64

7) Line 64. What do you mean by “natural” abundance? 

R= We have corrected the phrase and changed the word “natural” by “high” to avoid confusion L. 69

8) Suggested change: “The Solanaceae is…” 

R= the correction has been made 

9) Line 78. Introduce authority and year for eggplant as you do for achiote in line 85

R= authority and year have been included for eggplant L. 82

10) Line 79. Remove “thus” as self-compatibility does not imply self-pollination done

R= we have removed the word “thus”

11) Line 89. Which region is the major producer of achiote? 

R= Yucatan is the largest producer and consumer of achiote in Mexico, this Information is now included in L. 93-94

12) Line 94. AHB. I am not a fan of acronyms. Could you simply define and use “honeybees” throughout the paper? 

R=We could replace the acronyms but we would like to explain that we used them to save us from having to use 3 words across the whole document as we would like to emphasize that these are Africanized honey bees. We also think it may be more useful when presenting Figures and results. We hope that the referee would agree with our explanation to support keeping the acronyms

13) Line 94. No question has been formulated. Rephrase. 

R= As suggested we have rephrased this section L. 96-99

14) Line 116-117. More details needed. How many observers/cameras? Provide a standardised unit for observations (person-hours, etc.). 

R=We agree with the reviewer that this info is basic for assessment of the quality of our surveys. We have included these details in the paragraphs between L. 119,128 and L. 150,156.

15) Line 117. How was it determined if a bee buzz pollinated. Details needed as this is essential for the functional classification done in the next paragraph.

R= We agree with the reviewer, and have explained how this was determined in L. 125-128.

16) Line 119. Why was this initital analysis done? What was the sample size? It may be best to formally analyse this and report in the results with sample size details, etc. 

R= The sentence was rephrased. The data were the same as described in the materials and methods for the estimation of Frequency and behavior of floral visitors, there is no different data set. We first plotted the abundance of each of the genera present on each crop and from here we decided to separate the different visitors in different guilds to conduct further analyses L. 137-148.

17) Lines 121-122. How were these determinations done? Are voucher specimens collected and stored? How many species per genus? The information on determination appears briefly later. Consider re-organising.

R= We moved this info ahead for clarity L. 133-136.

18) Line 147. Was time nested in day? More details of statistical model are needed. Consider providing the full model used, and make sure it matches the results (eg. lines 233-236). You need to also explain whether the model assumptions are met.

R=Details of the statistical model are now presented in L. 172-179.

19) Line 152. “Both functional groups” I think you mean the three functional groups? Clarify. 

R= the correction has been made L. 144-145

20) Line 162. Why 2-3 visits? Was visit number included in the statistical model, and if not why not?

R=In this section we explain now that the analysis was used to statistically compare different treatments in each crop. As there were different treatments in accordance with the number of honey bee visits that individual flowers received, these were included in the model. We named these treatments as follows: AHB1 for flowers that received one visit of a honey bee, AHB2 for those that received two visits, and so on L. 190-197 and L. 234-244

21) Line 193. Not sure what the AHB2, AHB3,… mean. 

R= As explained above, AHB1 is the acronym used for flowers that were visited once by honey bees, AHB2 the one used for flowers that received two visits by honey bees, and so on. 

22) Line 198-200. Explain what the Friedman test is. 

R= We expanded the explanation of the Friedman test L. 238-244

23) Line 34-347. Please rephrase. Most genera in Solanaceae are not buzz pollinated. Within Solanum, not all species have anthers fused into a cone (the majority do not).

R= the sentence has been rephrased to include this observation made by the reviewer L. 449-450

24) Line 414. In this sentence, consider including in the discussion a more nuanced evaluation of the benefit of honeybees for “sustainable” pollination (as you state at the end of the paragraph). To the extent that honeybees impact native bees, it might not be very sustainable in a broader context. 

R= we have included a few lines to summarize the apparent effect of the interaction among honey bees and native bees L. 521-524.

Reviewer #2: The study provides some very interesting information about how non-native, feral honey bee colonies affect buzz-pollinated crops. Your analyses support your conclusions that 1) honey bees have become the dominant visitors to these crops, 2) that despite their lack of buzz pollination behavior honey bees seem to pollinate eggplant effectively by visiting multiple times, and 3) that honey bees may be decreasing pollination success for annato by competing with more effective native pollinators. These conclusions contribute interesting and useful information for anyone interested in agriculture or bee conservation in the tropics, and should be published. It was especially interesting to see how multiple visits by honey bees seem to compensate for their inability to vibrate the anthers of eggplant flowers but there is no similar effect in annato flowers. The experiments allowing 1, 2, or 3 honey bee visits and counting or estimating seed production are impressive. I imagine that took a lot of careful work.

R= Thank you for your positive comments and also for the time to revise our ms, with insightful remarks that are key for improving it. We reply to each of your queries below.

My main suggestion would be to double-check your bee abundance data for the 3 guilds and to make sure that you provide all data for figure 5. Table 1 does not match the data in your eggplant and annato data files. I have some suggestions below to add or clarify information at particular lines and figure/table captions.

R= We have checked the data again and try to include more details to clarify the presented information in general. We hope the changes are sufficient

You do not need to do this, but I would highly recommend providing your R code and providing your data in separate (ideally csv) files that can be read directly from your R code. This would help make your study more reproducible and easier to interpret. It would also help you to double-check your results (or clarify what data you included in each summary/analysis).

R=The csv files have been created and will be included as supplementary files

Introduction-

Good introduction, sets up the importance/questions well

R= Thank you for the comments, we have included some additional information on native bees L. 61-64.

Line 44-45: I know you understand that many crops are self-compatible or wind pollinated, but the way you say this, it makes it sound like all crops need visits from pollinators. 

R=We have rephrased this sentence L. 45

Materials and Methods-

Overall thorough description of your methods, but there are a few places that need clarification

Lines 116-117: Did you pause walking the transects each time you saw a bee and record it? 

R= We have included these details, we paused but the total time of observations was maintained Explained in L. 123-124 and 153-155

Line 154: When I hear the word proximate, I usually think of close in spatial location, but here you mean close in time, correct? 

R= yes, the word has been changed to avoid confusion

Line 198: So the data violated the assumption of normality and you used a non-parametric test instead?

 R= yes, this has been explained in more detail in L. 238-244

Line 218: I'm assuming the dotted line is this function? It looks like a good fit, but it would be helpful to include a little more detail about how you decided that this was the best fit. 

R= An explanation is now presented in L. 238-241.

Lines 219-228: It was helpful that you included a description of the formula here. R= Thanks

Results-

Overall thorough and clear, but there seems to be a problem in one of the tables and the captions need more detail

R= The observation made by the referee is correct, we have corrected table1 and we have included more detail in the captions of tables and figs.

Line 296: So am I correct that your point here is that even though the increase in seed production with unlimited access to the pollinators was statistically indistinguishable from the seed production produced by hand pollination, it's still possible that a different community of pollinators (maybe one with more native bee species) could have done a better job of providing pollination services to the eggplants?

R= We are expanding on this idea in L. 499-503

Line 303: Do you mean 38% less than the maximum possible? 

R= this figure has been corrected L. 406-407

Table 1:

There seems to be a few errors in the eggplant column and many errors in the annato column when I compare them to your data files. Please double-check these!

R= Yes, there were some inconsistencies with the data that have been updated- Table 1

What does EE refer to?

R= The acronym has been changed to the English for Standard Error (S.E.)- table 1

Picky comment- I would recommend presenting times of day in the format 8:00 or 16:00 rather than 8h and 16h because at first I thought those were durations

R= The format for time of day in Table 1 has been changed to avoid confusion

Figure 1:

This figure contains interesting information about the diversity of pollinators you found, but I wonder if it wouldn't be better to present pie charts because it would better convey the relative proportions. But that's personal preference.

R= We followed the reviewer’s suggestion and organized the data in pie charts to better appreciate patterns compared with the original Fig 1. Accordingly, we replaced the original bar graphs L. 167-170.

Figure 2:

It would help to put in the caption what the dotted line represents (maybe even including the equation you present in the text) because saying correlation makes me expect a straight line here (clearly you didn't just look for a linear correlation, but it would help to clarify that).

R= The caption has been changed and the equation added as the reviewer suggested L. 222-223

Figure 3:

This figure is very clear and contains a lot of information about the variation and trends

R=Thank you

Figure 4:

This figure is very helpful, providing context for the methods and discussion. Very nice photos of the anthers!

R= Thanks

Figure 5:

Why are all the y values between 0 and 3? Was that the range for both species or did you normalize somehow? 

R= The number represent the maximum number of fruit produced per treatment per plant. As there were three experimental flowers per treatment, the maximum number of fruit that could be produced is 3.

 I don't see where those numbers are in your eggplant data file.

R= we have included these data in the book called “Fruit produced per plant”

Table 2:

The percentages of the hand-pollinated results are very helpful. I think the letters to indicate significant differences would be good to add to the figure.

R= the letters indicating significant differences among treatments have been included in Figs 5 and 6

Table 3:

It would helpful to switch the NB and PLUS rows so they match the order in the figure

R= the rows of NB and PLUS in Table 3 have been changed to match the order in the Figure.

S1 Video:

Very interesting behaviors! You don't need to do this, but I wonder if it would make the video more likely to be used by others to either slow the it down or use video stabilization so it is easier to focus on the bee's behavior rather the flower moving in the wind (I know ffmpeg has a way to do this, although I haven't used it: https://www.paulirish.com/2021/video-stabilization-with-ffmpeg-and-vidstab/). If you do either of those things, please include a description of what you did in the video caption

R= an explanation of the use of the Wondershare Filmora 11 video editor to stabilize the video has been included in the video caption L 755-757.

S2 Video:

This looks like it was slowed down. It would be helpful to indicate how you did that (what % change, when in video).

R= an explanation of the use of the Wondershare Filmora 11 video editor and the percentage reduction in speed has been included in the video caption L 759-761.

S3 Video:

The cloud of pollen is quite striking. I'm not sure that you need the arrow, but if you keep it, I think it would be good to say in the caption that it is pointing to the cloud of pollen

R= The arrow has been removed from the video and the caption corrected L 763-766

Discussion-

It was helpful that you related your findings to previous literature on the pollination of buzz-pollinated crops and talked about potential reasons for the difference in efficiency of honey bees across the 2 crops. Your conclusions about supporting diverse native pollinators to maximize pollination services across crops seems reasonable.

R= Thanks for the comments.

Line 402-403: How are you calculating AHB contribution here? Is this because native non-sonicating bees were also poor pollinators per trip? R= AHB visit contributed an estimated 36-42% to the productivity of this crop in the field (comparing the contribution of AHB1 and AHB4 to OPEN). The explanation is now presented in more detail in L. 507-510

---

## [Editor Report · Decision Letter 1]

11 Jan 2023

Pollination service provided by honey bees to buzz-pollinated crops in the Neotropics

PONE-D-22-23499R1

Dear Dr. Quezada-Euan,

We’re pleased to inform you that your manuscript has been judged scientifically suitable for publication and will be formally accepted for publication once it meets all outstanding technical requirements.

Kind regards,

Olav Rueppell

Academic Editor

PLOS ONE
---

## [Editor Report · Acceptance letter]

16 Jan 2023

PONE-D-22-23499R1 

Pollination service provided by honey bees to buzz-pollinated crops in the Neotropics 

Dear Dr. Quezada-Euan:

I'm pleased to inform you that your manuscript has been deemed suitable for publication in PLOS ONE. Congratulations! Your manuscript is now with our production department. 

Kind regards, 

on behalf of

Dr. Olav Rueppell 

Academic Editor

PLOS ONE